# Derivation of Heat Conductivity from Temperature and Heat Flux Measurements in Soil

**Victor Stepanenko** [1,2,3,4,*], **Irina Repina** [1,3,4,5,†] **and Arseniy Artamonov** [4,5,†]

1. Research Computing Center, Lomonosov Moscow State University, 119234 Moscow, Russia; iar@ifaran.ru
2. Faculty of Geography, Lomonosov Moscow State University, 119234 Moscow, Russia
3. Moscow Center of Fundamental and Applied Mathematics, 119234 Moscow, Russia
4. Scientific and Educational Centre "Environmental Dynamics and Global Climate Change", Yugra State University, 628012 Khanty-Mansiysk, Russia; sailer@ifaran.ru
5. A.M.Obukhov Institute of Atmospheric Physics RAS, 119017 Moscow, Russia
* Correspondence: stepanen@srcc.msu.ru
† For Irina Repina and Arseniy Artamonov, the primary affiliation is No. 5.

**Abstract:** The general inverse problem formulation for a heat conductance equation is adopted for the types of measurement routinely carried out in the soil active layer. The problem solution delivers a constant thermal diffusivity coefficient $a_0$ (in general, different from true value $a$) and respective heat conductivity $\lambda_0$ for the layer, located between two temperature sensors and equipped with a temperature or heat flux sensor in the middle. We estimated the error of solution corresponding to systematic shifts in sensor readings and mislocation of sensors in the soil column. This estimation was carried out by a series of numerical experiments using boundary conditions from observations on Mukhrino wetland (Western Siberia, Russia), performed in summer, 2019. Numerical results were corroborated by analytical estimates of inverse problem solution sensitivity derived from classical Fourier law. The main finding states that heat conductivity error due to systematic shifts in temperature measurements become negligible when using long temperature series, whereas the relative error of $a$ is approximately twice the relative error of sensor depth. The error $a_0 - a$ induced by heat flux plate displacement from expected depth is 3–5 times less than the same displacement of thermometers, which makes the requirements for heat flux installation less rigid. However, the relative errors of heat flux observation typical for modern sensors ($\pm15\%$) cause the uncertainty of $a$ above 15% in absolute value. Comparison of the inverse problem solution to $a$ estimated from in situ moss sampling on Mukhrino wetland proves the feasibility of the method and corroborates the conclusions of the error sensitivity study.

**Keywords:** moss layer; heat conduction; inverse problems; measurement errors



## 1. Introduction

The coefficient of heat conductivity is a thermal property of soils that is important for correct representation of soil–atmosphere interactions in numerical weather prediction and Earth system models. A variety of parameterizations have been proposed for heat conductivity [1], which have different accuracies for different types of soil and soil states. Thus, robust measurement techniques are needed for further improvement of such models.

The coefficient of heat conductivity $\lambda$ for any material is defined in the phenomenological Fourier law for heat flux:

$$q = -\lambda \frac{\partial T}{\partial z}, \tag{1}$$

where $q$ is the heat flux, $T$ is temperature, and $z$ is spatial coordinate parallel to temperature gradient.

For multicomponent environments, Equation (1) is also used, but $\lambda$ is now the effective thermal conductivity of the medium, which depends on its structure and thermal

conductivity of its components [2–4]. Since the arrangement of the medium components is usually chaotic, and the surfaces separating them are extremely complex, the method of two-sided estimates is used for calculating the thermal conductivity [5]. In this method, the medium is considered as a set of flat, regularly alternating isotropic layers that fill its entire volume. Thermal conductivity may be calculated for heat flux along the layers (Voigt model), which provides the upper estimate for $\lambda$, and across the layers (Reuss model), which is the lower estimate. For the real structure of the medium, a combination of these models is applied. This approach has been used to calculate the thermal conductivity of snow [6,7] and moss [8]. A similar approach, but taking into account the larger number of medium components (solid particles, water and air), is typically used to compute the thermal conductivity of soil [1,9]. This technique, in particular, facilitates the study of porosity and liquid water saturation effect on the thermal conductivity of various media.

Direct measurement of the thermal conductivity coefficient of natural objects (soils, rocks, vegetation, snow, etc.) can be carried out both in the field and in laboratory conditions [10–15]. In field conditions, the thermal conductivity coefficient may be determined from the measured heat flux and temperature gradient in the material [16]. Laboratory methods use special equipment and are divided into stationary and non-stationary methods. Stationary methods are based directly on Fourier's law, and the heat flux through the sample over time reaches a steady state; in non-stationary methods, the heat flux does not reach a constant value.

Stationary methods may use either an absolute (direct) or relative approach. In direct measurements, thermal conductivity is calculated directly from the experimentally found values. Relative methods require a reference material with a known thermal conductivity. Of the absolute methods, the controlled hot plate method [17] and the cylinder method [18] are the most applicable. Relative methods include the heat-flux meter method [19], the direct heating method (Kohlrausch method) [20] and hot wire method [21].

Non-stationary methods allow for direct measurement of the thermal conductivity. These include the frequency division method [22] and the laser flash method [23]. In work [24], the advantage of non-stationary methods for laboratory studies of soils and grounds, in particular, using thermal needle probes, is shown, since in addition to the shortening of the study (no wait for a stationary thermal regime is required), the sample does not lose moisture during the study.

Direct measurements of heat conductivity mentioned above require deployment of additional sensors in soils or transport of samples to a lab, whereas efficient use of conventional devices such as temperature sensors and heat-flux plates for estimation of $\lambda$, would substantially increase the number of sites, where soil heat conductivity is assessed.

This paper formulates a generalized method for obtaining soil thermal conductivity as a solution of the inverse problem. It extends the method which uses three-level temperature measurements developed in [25–27] by considering a case, where heat-flux observations are available. The sensitivity of the inverse problem solution to input data errors is studied, and the accuracy of the method is compared to that of the traditional Fourier approach and contemporary thermal conductivity sensors. Finally, an inverse problem method is verified using data of measurements in a water-saturated moss layer in Western Siberia.

## 2. Materials and Methods

Consider a heat conductance problem in a medium which has homogeneous spatial distribution of thermodynamic properties:

$$T_t = aT_{zz}, \ z \in (z_1, z_2), \ t \in (0, t_1), \tag{2}$$

$$B_1 T|_{z=z_1} = f_1(t),$$

$$B_2 T|_{z=z_2} = f_2(t), \tag{3}$$

$$T|_{t=0} = T_0(z), \tag{4}$$

where $a = \lambda/(\rho c) > 0$ is a thermal diffusivity coefficient, $\rho$ is density, $c$ is specific heat capacity (all are constants), $B_{1,2}$ are differential operators of boundary conditions (hereafter, we assume $B_{1,2} = I$, with $I$ standing for unity operator, implying Dirichlet boundary conditions), bottom indices $z$ and $t$ denote derivatives on time and depth. If $a$ is known, then a problem of finding a function $T(z, t)$ satisfying equation (2), boundary (3) and initial (4) conditions is called a direct problem. However, very often $a$ is unknown, whereas additional information on solution $T(z, t)$ is available, e.g., measurements of temperature at an intermediate depth $z_3 \in (z_1, z_2)$ expressed by a function $T_m(z_3, t) = f_3(t)$. In this case, one may pose a problem of seeking $a$ given this additional constraint; this classical kind of problem is called *an inverse problem*. Specifically, the solution of inverse problem is

$$a_0 = \arg\min_{a'} ||T_{a'} - T_m||^2, \tag{5}$$

where $T_{a'}$ is a solution of (2)–(4) under given $a = a'$, $|| \cdot ||$ is an appropriately chosen norm, and $T_m(z, t)$ is an observed temperature field. In the following, we assume that the volumetric heat capacity $\rho c$ is known, so that for each value of temperature diffusivity, $a$, the heat conductivity $\lambda = a \rho c$ is found automatically. The norm in (5) may be defined in different ways, depending on which temperature data are available. In a more general case, the objective function to be minimized on $a'$ may be defined as a sum of norms containing different functions of $T_{a'}$, which are measured, say, heat flux. For instance, a sensor of temperature is deployed at depth $z_3$, and a heat flux sensor is installed at depth $z_4 \in (z_1, z_2)$. In this case, the objective (loss) function to be minimized is a sum of respective norms:

$$\begin{aligned} \Phi(a') = & C_1 \int_0^{t_1} [T_{a'}(z_3, t) - T_m(z_3, t)]^2 dt + \\ & C_2 \int_0^{t_1} [a'(T_{a'})_z(z_4, t) - F_m(z_4, t)]^2 dt. \end{aligned} \tag{6}$$

Here, constants $C_1$ and $C_2$ are non-negative weights which may be chosen as inversely proportional to standard errors of respective measurements, and $F_m$ is the measured heat flux. In (6), the temperature difference norm is summed up with a norm of heat flux difference. The two typical measurement settings used in soil active layer monitoring lead to the following inverse problem specifications:

- $C_1 > 0$, $C_2 = 0$; the data of three temperature loggers at different depths are used ($\Phi(a')$ is RMSE of temperature squared) (this method coincides with that used in [25–27]), hereafter referred to as TEMP method (hereafter, the depth of measurements for this setting is denoted as $z_3 = z_m$);
- $C_1 = 0$, $C_2 > 0$; the data of two temperature sensors and a heat flux plate located in between are used ($\Phi(a')$ is the RMSE of heat flux squared), hereafter referred to as the FLUX method (hereafter, the depth of measurements for this setting is denoted as $z_4 = z_m$).

Calculation of $\Phi(a')$ is performed after solving the direct numerical problem (2)–(4) at a given $a'$, and minimization of $\Phi(a')$ may be attained by a number of methods e.g., Monte-Carlo simulations [28] or gradient descent method. The latter is used in this study in Barzilai–Borwein modification [29]. As both $T_m$ and $F_m$ are measured with error, the sensitivity of $a_0$ to those errors must be assessed. For reference, we compare $a_0$ obtained by the method described above to estimates from the classical Fourier solution:

$$a_0 = \frac{\pi(z_2 - z_1)^2}{T_d \log^2(A_{z_1}/A_{z_2})}, \tag{7}$$

($T_d = 24\,\text{h}$ and $A_{z_i}$, $i = 1, 2$ are diurnal temperature magnitudes at the top and bottom of the layer considered), and the accuracy of heat conductivity sensors available on the market.

The measurements of soil temperature have been conducted on Mukhrino bog station in Western Siberia in summer 6–20 June 2019, (Figure 1). The sensors were placed in the

*Sphagnum* moss layer 5–25 cm below surface, with the entire layer located beneath the water level. Davis Instruments stainless steel temperature probes with two-wire termination were deployed at depths 5, 15, and 25 cm. The advantage of this experimental setup for testing the method of thermal conductivity derivation (in TEMP configuration) is that the "true" temperature conductivity of the moss layer is known to a high-accuracy: water constituted about 97.5% of the layer by mass, whereas the almost permanently stable stratification prevented convective motions [30], so that the thermal conductivity of the layer is close to molecular conductivity of water, $a \approx a_w$ (see a more exact estimate below in this section).

We performed two tests of the method proposed. In the first test, the measured temperature series at 5 and 25 cm are used as boundary conditions to compute temperature and heat flux series at 15 cm by solving (2)–(4) under $a = a_w$ (the PDE solver uses the scheme, which is central-difference in space and implicit in time). Thus produced series at 15 cm are then used as true solution (or "measured" series) to solve the inverse problem for *a* with the same top and bottom boundary conditions as in a direct problem. In specification of $\Phi(a')$, (6), temperature series in TEMP settings are used in a 1 min time step, and the heat flux in the FLUX problem is given with a 30 min time step (i.e., averaged inside consecutive 30-min intervals of "true" solution); the latter is a typical averaging interval of observed heat flux in soil monitoring practice. Thus, the accuracy of the method is estimated.

The sensitivity of the method accuracy to the error of temperature (at all three depths) and heat flux (in the middle of layer) series was assessed in this first test. The error for temperature is given by Gaussian noise that is uncorrelated in time with mean $\pm 0.1$ °C (typical for conventional temperature sensors) and $\sigma = 0.1$ °C ; assuming mean error values $-0.1$ °C, 0 °C, $+0.1$ °C at each of three levels provides 27 combinations, some of which are equivalent. For heat flux, we assume a constant relative error $\pm 15\%$, which can be seen as an upper estimate for the modern heat-flux plates (see, e.g., [31]). In addition, the error of sensors' positions is introduced to sensitivity analysis, assuming they are $\pm 1$ cm for each temperature sensor, again providing 27 combinations; the effect of the same error in the level of heat flux sensor deployment is estimated as well.

In the second test, the measured temperature series at 5 cm, 15 cm and 25 cm of moss layer of Mukhrino bog are used to solve the TEMP type of inverse problem (Figure 2). The difference to the first test is that for a 15 cm depth, *measured* temperature series are used (instead of series precomputed with "true" temperature diffusivity $a = a_w$), and thus the resulting $a_0$ value is compared to *a* estimated for this water-saturated moss layer given the measured water and organics mass fractions (see below). The similar test of the FLUX-type inverse problem solution was not carried out, as we did not have reliable *observed* heat flux data from the field campaign.

On the Mukhrino field station, we sampled the top of the moss layer to obtain the values of water and organics mass fractions, $M_w$ and $M_{org}$, respectively ($M_w + M_{org} = 1$). Given these parameters, the moss layer porosity *p*, the volumetric heat capacity $c_{vol}$, heat conductivity coefficient $\lambda_s$, and thermal conductivity coefficient *a*, are:

$$p = \frac{\gamma}{1 + \gamma}, \; \gamma \doteq \frac{\rho_{org} M_w}{\rho_w M_{org}}, \tag{8}$$

$$\lambda_s = \lambda_w^p \lambda_{org}^{1-p}, \tag{9}$$

$$\rho_s = p \rho_{w0} M_w^{-1}, \tag{10}$$

$$c_{vol} = \rho_s \left( M_w c_w + M_{org} c_{org} \right), \tag{11}$$

$$a = \frac{\lambda_s}{c_{vol}}. \tag{12}$$

Given the measured value $M_w = 0.975$, and the values of other parameters $c_{org} = 2250$ J/(kg·K) [32–34], $c_w = 3990$ J/(kg·K), $\rho_{org} = 1300$ kg/m$^3$, $\rho_w = 1000$ kg/m$^3$, $\lambda_{org} = 0.3$ W/(m·K) [33], $\lambda_w = 0.561$ W/(m·K), we obtain $ac_{vol} = 0.554$ W/(m·K). Here, the widely accepted geometrical mean is used to estimate the heat conductivity $\lambda_s$ of water-saturated soil.

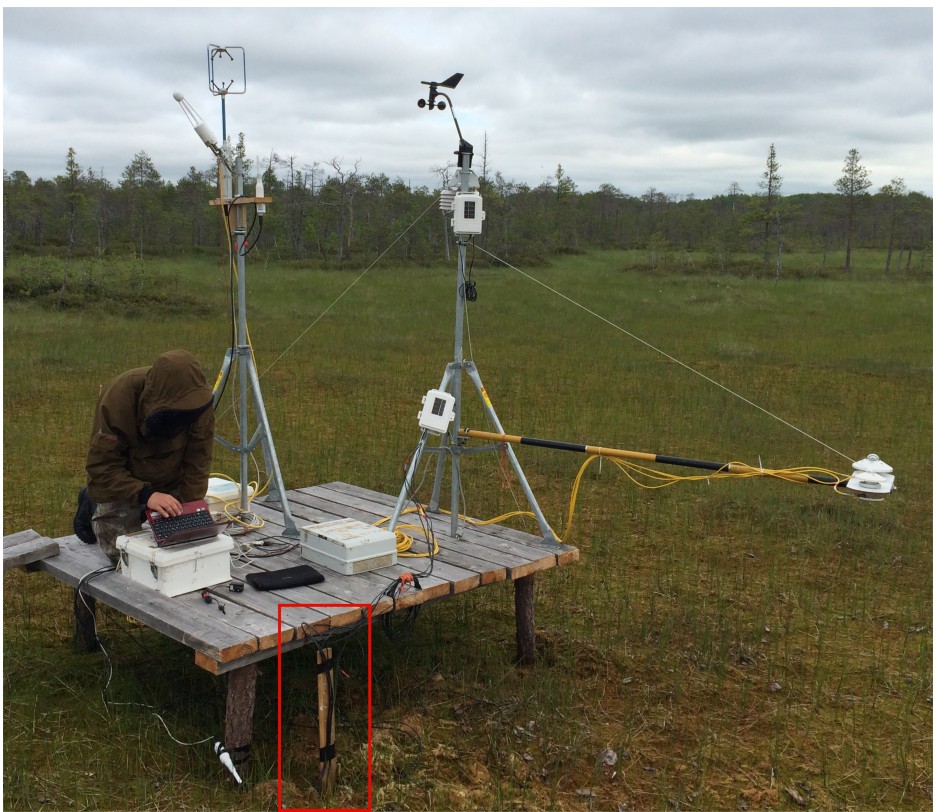

**Figure 1.** The landscape view and setup of measurements on Mukhrino bog (Khanty–Mansiysk region, Russia) in summer, 2019. The red rectangle shows the vertical string of temperature and heat flux sensors in the moss layer.

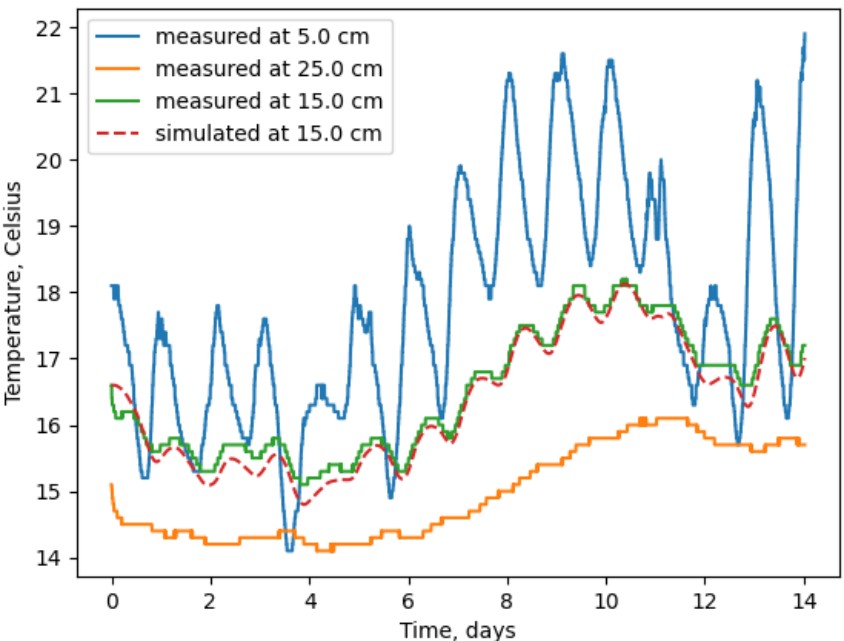

**Figure 2.** Moss temperature series measured at top (5 cm), bottom (25 cm) and middle (15 cm), of the test moss layer, and simulated in the middle of the layer at the optimal value of the temperature diffusivity coefficient $a_0$.

## 3. Results and Discussion

We start with the first (synthetic) test of the inverse problem method. The error of the heat conduction coefficient found by solving the inverse problem formulated in the

previous section significantly depends on the type and magnitude of the input data error (Table 1). In the table, of all the combinations of input data systematic errors, only those with the largest impact on inverse problem solution are shown. In addition, for each series of experiments, the $a_0$ values using zero mean errors are given. The loss function in this case attains the level of stochastic error magnitude of temperature or heat flux, respectively; the solution $a_0$ is 0.2–0.5% different from the reference value $a_w$. The latter deviations are partly due to inaccuracy of the numerical minimization procedure.

According to Table 1, the smallest uncertainty of $a_0$ is caused by systematic shifts in measured temperature in the TEMP method, not exceeding 3% by absolute value. The sign of error here coincides with that of $\delta T_m - \delta T_1$, implying that the temperature error at the bottom boundary is less important compared to uncertainties at two other levels. In the FLUX method, the uncertainty of $\pm 15\%$ in heat flux measurements causes the deviation of $a_0$ from "true" value $a_w$ up to $\pm 20\%$, where $a_0 - a_w$ has the same sign as the heat flux relative error $\delta F_m$; at a given $\delta F_m$ choosing different combinations of top and bottom temperature errors does not considerably change the deviation $a_0 - a_w$. Thus, the FLUX method is much less precise, than the TEMP method, if considering only the accuracy of sensor readings. However, the sensitivity of solution $a_0$ to sensor deployment errors of $\pm 1$ cm is much larger for the TEMP method, reaching more than 40% compared to less than 15% in the FLUX method. In both methods, the error sign is the same as that of $\delta z_m - \delta z_1$, again suggesting that observation uncertainties at the middle and top levels have more impact on solution $a_0$ compared to those at the bottom.

**Table 1.** Inverse problem solution as a function of input data errors. $\delta(\cdot)$ is a mean error of variable $(\cdot)$. True solution is $a\rho c = a_w \rho c = 0.561$ Wm$^{-1}$K$^{-1}$. The solutions with zero input data errors and with maximal $|a_0 - a_w|$ are shown.

| Solution under different temperature mean errors (TEMP setup) | | | | |
|---|---|---|---|---|
| $\delta T_1$, K | $\delta T_m$, K | $\delta T_2$, K | $a_0 \rho c$, Wm$^{-1}$ K$^{-1}$ (relative error) | RMSE, °C |
| 0 | 0 | 0 | 0.559 ($-0.4\%$) | 0.10 |
| $-0.1$ | 0.1 | $-0.1$ | 0.570 ($+1.6\%$) | 0.23 |
| $-0.1$ | 0.1 | 0 | 0.567 ($+1.1\%$) | 0.18 |
| 0 | $-0.1$ | 0.1 | 0.553 ($-1.4\%$) | 0.18 |
| 0 | 0.1 | $-0.1$ | 0.571 ($+1.8\%$) | 0.18 |
| 0.1 | $-0.1$ | $-0.1$ | 0.558 ($-0.5\%$) | 0.15 |
| 0.1 | $-0.1$ | 0 | 0.546 ($-2.7\%$) | 0.18 |
| 0.1 | $-0.1$ | 0.1 | 0.546 ($-2.7\%$) | 0.22 |
| Solution under different temperature sensor depth errors (TEMP setup) | | | | |
| $\delta z_1$, cm | $\delta z_m$, cm | $\delta z_2$, cm | $a_0 \rho c$, Wm$^{-1}$ K$^{-1}$ (relative error) | RMSE, °C |
| 0 | 0 | 0 | 0.564 ($+0.5\%$) | 0.10 |
| $-1$ | $+1$ | $-1$ | 0.787 ($+40.3\%$) | 0.31 |
| $-1$ | $+1$ | 0 | 0.801 ($+42.8\%$) | 0.23 |
| 0 | $-1$ | $+1$ | 0.441 ($-21.4\%$) | 0.23 |

**Table 1.** *Cont.*

| 0 | +1 | −1 | 0.655 (+16.8%) | 0.24 |
|---|---|---|---|---|
| +1 | −1 | 0 | 0.366 (−34.8%) | 0.23 |
| +1 | −1 | +1 | 0.357 (−36.3%) | 0.30 |
| **Solution under different temperature and heat flux mean errors (FLUX setup)** | | | | |
| $\delta T_1$, K | $\delta F_m$, % | $\delta T_2$, K | $a_0\rho c$, Wm$^{-1}$ K$^{-1}$ (relative error) | RMSE, W m$^{-2}$ |
| 0 | 0 | 0 | 0.563 (+0.4%) | 0.99 |
| −0.1 | +15 | 0 | 0.654 (+16.6%) | 1.11 |
| −0.1 | +15 | +0.1 | 0.671 (+19.6%) | 1.16 |
| 0 | −15 | −0.1 | 0.467 (−16.8%) | 1.06 |
| 0 | +15 | +0.1 | 0.654 (+16.6%) | 1.11 |
| +0.1 | −15 | −0.1 | 0.456 (−18.7%) | 1.13 |
| +0.1 | +15 | +0.1 | 0.638 (+13.7%) | 1.08 |
| **Solution under different heat flux plate depth errors (FLUX setup)** | | | | |
| $\delta z_1$, cm | $\delta z_m$ cm | $\delta z_2$, cm | $a_0\rho c$, Wm$^{-1}$ K$^{-1}$ (relative error) | RMSE, W m$^{-2}$ |
| 0 | 0 | 0 | 0.560 (−0.2%) | 1.01 |
| −1 | 0 | −1 | 0.574 (+2.3%) | 1.16 |
| −1 | +1 | −1 | 0.586 (+4.5%) | 1.38 |
| −1 | +1 | 0 | 0.616 (+9.8%) | 1.34 |
| −1 | +1 | +1 | 0.641 (+14.3%) | 1.22 |
| 0 | −1 | +1 | 0.560 (−0.2%) | 1.26 |
| +1 | −1 | −1 | 0.481 (−14.3%) | 1.31 |
| +1 | −1 | 0 | 0.499 (−11.1%) | 1.41 |
| +1 | −1 | +1 | 0.513 (−8.6%) | 1.55 |

For interpretation of errors $a_0 - a_w$ obtained in numerical simulations, the standard Fourier solution ("Fourier law") for the heat conduction problem provides a convenient framework, because it is tractable analytically. The errors of inverse problem solution due to sensor inaccuracies and vertical displacement are (see Appendix A) as follows:

$$\delta a \approx 0, \ \delta T_m \neq 0, \ \delta z_m = 0, \tag{13}$$

$$\frac{\delta a}{a} = 2\frac{\delta z_m}{z_m}, \ \delta T_m = 0, \ \delta z_m \neq 0, \tag{14}$$

for TEMP setting and

$$\frac{\delta a}{a} \approx \frac{\delta F_m}{100} \frac{2(1 + \Delta z_m/z_*)}{(1 + \Delta z_m/z_*)^2 + (\Delta z_m/z_*)^2}, \ \delta F_m \neq 0, \ \delta z_m = 0, \tag{15}$$

$$\frac{\delta a}{a} \approx \frac{\delta z_m}{z_*} \frac{2(1 + \Delta z_m/z_*)}{(1 + \Delta z_m/z_*)^2 + (\Delta z_m/z_*)^2}, \ \delta F_m = 0, \ \delta z_m \neq 0. \tag{16}$$

for FLUX setting, where $\delta a = a_0 - a$, $\Delta z_m = z_m - z_1$, and $z_* = \sqrt{\frac{aT}{\pi}}$ (*T*—period of temperature oscillations). Note that the change of sign has been made before $\delta z_m$ to conform with different displacement definitions used in numerical experiments of Table 1 compared to derivation in Appendix A.

Strictly speaking, estimates of uncertainty from Fourier law are not directly comparable to uncertainties of numerical solution of the inverse problem as the Fourier solution assumes a different lower boundary condition and single-harmonic oscillation at the upper boundary; still these estimates well conform with the numerical results in Table 1. Indeed, the small values of uncertainty due to systematic temperature measurement errors in the TEMP problem setup agree with $\delta a \approx 0$ from Fourier theory and are anticipated to reduce with extension of the time period studied[1]. Then, from Fourier theory, for the sensor depth error $\delta z_m - \delta z_1 = 0.1,\ 0.2$ m, we obtain the relative error of 20% and 40% in $a_w$, respectively, which is again in accordance with Table 1.

For the FLUX inverse problem, in analytical estimates, we have dependence of $\delta a / a$ on period $T$, which complicates the analysis. In Figure 2, one can see two pronounced periods comprising temperature series used as boundary conditions: the diurnal period and the synoptic period—the latter can be estimated as 12 days. For these two periods and $\delta F = 15\%$, the above analytical estimates provide $\delta a / a = 8.3\%$ and 18.6%, respectively; the corresponding value in Table 1 (for $\delta T_1 = 0$ K) is 16.6%, and it resides in the interval between these two values. The uncertainties of $\delta a / a$ due to flux sensor depth displacement presented in the table are not well reproduced by Fourier theory, as they demonstrate a strong influence of the lower boundary, which is not included in Fourier law. Nevertheless, the sign of the error is correctly predicted by the formula (16), which also provides values in a range from numerical simulations.

Now turn to the second test ot the method, where the computed optimal $a_0$ was compared to in-situ estimated thermal conductivity. The optimal value $a_0 c_{vol} = 0.514$ W/(m·K), is $-7.2\%$ from 0.554 W/(m·K), a reference value, estimated from moss sampling (see Section 2). There are at least two possible reasons for this discrepancy. First, the wooden rod used to deploy temperature sensors disturbed the natural moss medium and affected heat transport; this effect is not straightforward to quantify; however, one may notice that the heat conductivity of dry wood is about 2 times less than that of water, i.e., 0.2–0.3 W/(m·K) [35], which means it could contribute to reduction of apparent $a$, obtained from data of sensors attached to this rod. The second reason is the error in sensor depth. Setting the middle thermistor depth error as $\delta z_2 = 3$ and 4 mm provides an inverse problem solution $a_0 c_{vol} = 0.539$ W/(m·K) ($-2.7\%$) and $a_0 c_{vol} = 0.567$ W/(m·K) ($+2.3\%$), suggesting that the thermistor might have been placed 3–4 mm below the expected depth in the moss layer. This estimate is realistic given the technical procedure of the sensors' deployment in our field experiment. The temperature series corresponding to these optimal values are shifted from measured data by 0.1–0.2 K (Figure 2), which can be explained by the mean thermistor error.

Using the Fourier formula (7), we obtain the daily values of $a_0 c_{vol}$ from Mukhrino bog data, which have a mean over 14 days 1.03 W/(m·K) (+85.92%) with root mean square deviation 0.58 W/(m·K). The reason for this large error is a proximity of the diurnal magnitude of temperature variations at depth 25 cm to the measurement accuracy of 0.1 °C. Moreover, at days 6–12, the diurnal cycle is not discernible at a background of synoptic trend.

Given the measurement accuracy of modern thermal needle systems $\pm 5\%$ [36] for the heat conductivity coefficient, we may note the following. Using the TEMP inverse problem solution provides better accuracy for $a$ if the sensors are located exactly at the expected depths; this requires much care during the deployment procedure, because the TEMP method is very sensitive to depth mislocation. For the FLUX method, errors of modern heat flux plates must be reduced by about 3 times to obtain the same accuracy as thermal needle systems provide; the high accuracy in terms of the depth of the sensor is not as important as in the TEMP method case.

This work proposes a generalized inverse problem formulation for heat transfer in soils not undergoing phase changes. A special case of this problem statement, called TEMP

---

[1] According to Fourier solution, $a$ is a function of a ratio of temperature magnitudes at two depths; systematic temperature shifts do not change magnitudes and hence $a$.

configuration in this paper, was applied in the previous works [25–27,37] to northern soils. We propose to use heat flux plate measurements in between two temperature sensors as an alternative setting for estimation of the optimal thermal diffusivity. In addition, for the first time, we numerically and analytically estimate the inverse solution uncertainty due to errors of input data, which are Gaussian noise and systematic errors in temperature and heat flux observations as well as the shifts in deployment depths. The future derivations of thermal conductivity using the same instrumental settings may be now accompanied by similar uncertainty estimates.

## 4. Conclusions

In this study, the general inverse problem formulation for the heat conductance equation is adopted for the types of measurement routinely carried out in the soil active layer. The problem solution delivers a constant thermal diffusivity coefficient $a_0$ (in general, different from true value $a$) and respective heat conductivity $\lambda_0$ for a layer, which was located between two temperature sensors and equipped with a temperature or heat flux sensor in the middle. Given that the inverse problem solution may be sensitive to the uncertainty of input data, we estimated the error of a solution corresponding to systematic shifts in sensor readings and mislocation of sensors in the soil column. This estimation was carried out by a series of numerical experiments using boundary conditions from observations on Mukhrino wetland (Western Siberia, Russia), performed in summer, 2019. Numerical results were corroborated by analytical estimates of inverse problem solution sensitivity derived from classical Fourier law. The main finding states that heat conductivity errors due to systematic shifts in temperature measurements become negligible when using long temperature series, whereas the relative error of $a$ is approximately twice the relative error of sensor depth (for example, 20% error in $a$ takes place for sensor displacement of 1 cm around 10 cm expected depth). The error $a_0 - a$ induced by heat flux plate displacement from the expected depth is 3–5 times less than the same displacement of thermometers, which makes the requirements for heat flux installation less rigid. However, the relative errors of heat flux observation typical for modern sensors ($\pm15\%$) cause the uncertainty of $a$ above 15% in absolute value. Comparison of the inverse problem solution to $a$ estimated from in situ moss sampling on Mukhrino wetland proves the feasibility of the method and corroborates the conclusions of error sensitivity study.

**Author Contributions:** Conceptualization, V.S. and I.R.; methodology, V.S.; resources, I.R. and A.A.; writing—original draft preparation, V.S. and I.R.; writing—review and editing, I.R.; project administration, I.R.; measurements, A.A. All authors have read and agreed to the published version of the manuscript.

**Funding:** The work is supported by Russian Foundation for Basic Research, grants no. 18-05-60126 (temperature and heat flux measurements in moss layer) and no. 20-05-00773 (the method of inverse problem solution for soil heat conductivity), Russian Ministry of Science and Higher Education, agreement No. 075-15-2019-1621 (sensitivity of inverse problem solution to errors of input data), and the grant of the Tyumen region Government in accordance with the Program of the West Siberian Interregional Scientific and Educational Center (National Project "Nauka") (logistic organization of field work).

**Institutional Review Board Statement:** Not applicable.

**Informed Consent Statement:** Not applicable.

**Data Availability Statement:** The data presented in this study are available on request from the corresponding author.

**Acknowledgments:** The authors are grateful to E. Lapshina, A. Dmitrichenko and E.Dyukarev for providing conditions of fruitful work on Mukhrino station.

**Conflicts of Interest:** The authors declare no conflict of interest.

**Appendix A. Errors of Inverse Solution according to Fourier Law**

The Fourier solution for temperature vanishing $T_a$ at infinite depth is

$$T_a(z, t) = A_0 \exp(-z/z_*) \cos[\omega(t - \tau)], \tag{A1}$$

$$\tau = \frac{z}{2}\sqrt{\frac{T}{\pi a}}, \tag{A2}$$

$$z_* = \sqrt{\frac{aT}{\pi}}, \tag{A3}$$

$$\omega = \frac{2\pi}{T}, \tag{A4}$$

where $T$ is the period, $A_0$ is magnitude of temperature oscillations at the top of the soil domain considered. Assume we use the TEMP setting of the inverse problem, the true temperature obeys Fourier law; however, the temperature sensor has a constant error $\delta T$ and is deployed at the expected depth $z$ with displacement $\delta z$. In this case, the loss function (6) takes the form:

$$\int_0^{t_1} [\{T_a(z + \delta z, t) + \delta T\} - T_{a'}(z, t)]^2 dt \to \min_{a'} \Phi(a'). \tag{A5}$$

This minimization problem is solvable analytically, with solution $a' = a_0$. In the following development, we assume for simplicity that $t_1 \gg T$. This yields for $\delta a = a_0 - a$:

$$\delta a \approx 0, \; \delta T \neq 0, \; \delta z = 0, \tag{A6}$$

$$\frac{\delta a}{a} = -2\frac{\delta z}{z}, \; \delta T = 0, \; \delta z \neq 0, \tag{A7}$$

with $\delta a$ standing for absolute error $a_0 - a$. The first of these expressions is remarkably simple and can be used as a "rule of thumb" for assessing the uncertainty of the TEMP inverse problem solution.

In FLUX inverse problem formulation, the loss function including measurement errors takes a form:

$$\int_0^{t_1} [a(T_a)_z(z + \delta z, t)(1 + \delta F) - a'(T_{a'})_z(z, t)]^2 dt \to \min_{a'} \Phi(a'), \tag{A8}$$

with $\delta F$ being a relative error of flux observations. Analytically evaluating a value $a_0$ delivering the minimum to the above objective function provides

$$\frac{\delta a}{a} \approx \delta F \frac{2(1 + z/z_*)}{(1 + z/z_*)^2 + (z/z_*)^2}, \; \delta F \neq 0, \; \delta z = 0, \tag{A9}$$

$$\frac{\delta a}{a} \approx -\frac{\delta z}{z_*} \frac{2(1 + z/z_*)}{(1 + z/z_*)^2 + (z/z_*)^2}, \; \delta F = 0, \; \delta z \neq 0. \tag{A10}$$

Contrary to the TEMP problem, in FLUX, the relative solution error does not depend only on the relative error of measurements, but also on true value $a$, period $T$ and depth $z$.

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
