# Peer review of "Derivation of Heat Conductivity from Temperature and Heat Flux Measurements in Soil"

_land, doi:10.3390/land10060552_

Round 1

Reviewer 1 Report

The work is very important for the understanding of the soil thermal conductivity, especially in studies related to the surface energy balance. Regarding the manuscript I have some doubts/suggestions:

1. In line 27, “where q - the heat flux, T - temperature,…” perhaps leave “where q is the heat flux, T is the temperature,…”;

2. In the “Materials and Methods” section, only summer 2019 is mentioned, if possible indicating the days of the year (DOY).

3. If possible, mention the model of the used sensors, not only the manufacturer.

4. Also, in the “Materials and Methods” section, only temperature sensors are mentioned, however, information about heat flux sensors is mentioned throughout the text. If they were actually used, it is important to inform the sensors used (manufacturer, model, for example). If heat flux sensors have not been used, it is important to better describe the procedure adopted. The way of the manuscript it is currently written does not allow a clear understanding of this information and the procedure adopted. In another words, it is important to better describe these procedures.

5. In the text, it is mentioned that the sensors were installed at 5, 15 and 25 cm, in the figure, there is 15.5 cm, it is known of the small variation in the installation, however, it is important to standardize the information presented.

6. On line 110, I believe it is RMSE and not RMSD.

7. The information present in lines 159 to 161 (section “Results and Discussions”), relative to the characteristics of the soil, could be described previously in the section “Material and Methods”, next to line 122.

8. In Figure 1, it is possible to verify the use of a wooden bar to support the sensors, does it accompany the sensors to the installation depth? In this case, does it not distort the data? If possible, explain better the installation and the data treatment.

9. In Figure 2, suggestion, use dashed line for the simulated case, facilitating the distinction between measured and simulated data.

10. Table 1 could be better used (it seems to me that the data presented were under-analyzed, or better, superficially analyzed).

Reviewer 2 Report

This article considers a method to obtain the thermal conductivity of the soil, using three-level temperature measurements.

I think this paper is novel and very interesting for publication in “Land”. It has a sufficient impact and it represents an advance in the subject matter. The expected scientific impact is medium-high.

The Introduction is correct, with a notable bibliographic review. The methodology and work development seems correct, in general. The conclusions are supported by the results and are consistent with the aims set.

I think the paper is suitable for publication in “Land” with moderate revision.

I suggest the following changes:

1) The keywords should not be contained in the title, to improve the performance of search engines. So please change "soil”, “heat conductivity”, “temperature”, “heat flux” by another keywords.

2) In my opinion, it is necessary to express better, more clearly, what is the objective of the work, in the final part of the Introduction.

3) Line 89: “Sphagnum” must be written in italics.

4) Figure 2: Please remove the title from the top of the graph, as the same is expressed in the figure caption.

5) The main drawback that I see with the work is that it does not have a proper and consistent discussion. Please, you should compare your results with those obtained by other authors, so it is necessary to increase the number of bibliographic citations in this section. In this way we will be able to appreciate, as it seems, that his work really represents an interesting scientific advance.

Round 2

Reviewer 1 Report

The indicated suggestions were made.

Reviewer 2 Report

I consider that the authors have made the changes suggested in the paper, and I consider that this new version is sufficiently improved.